# Wnt Pathway: An Integral Hub for Developmental and Oncogenic Signaling Networks

**DOI:** 10.3390/ijms21218018

**Published:** 2020-10-28

**Authors:** Monica Sharma, Kevin Pruitt

**Affiliations:** Department of Immunology and Molecular Microbiology, Texas Tech University Health Sciences Center, Lubbock, TX 79430, USA; monica.sharma@ttuhsc.edu

**Keywords:** Wnt signaling, human diseases, cancer, antitumor immunity, cancer stemness, therapeutic strategies

## Abstract

The Wnt pathway is an integral cell-to-cell signaling hub which regulates crucial development processes and maintenance of tissue homeostasis by coordinating cell proliferation, differentiation, cell polarity, cell movement, and stem cell renewal. When dysregulated, it is associated with various developmental diseases, fibrosis, and tumorigenesis. We now better appreciate the complexity and crosstalk of the Wnt pathway with other signaling cascades. Emerging roles of the Wnt signaling in the cancer stem cell niche and drug resistance have led to development of therapeutics specifically targeting various Wnt components, with some agents currently in clinical trials. This review highlights historical and recent findings on key mediators of Wnt signaling and how they impact antitumor immunity and maintenance of cancer stem cells. This review also examines current therapeutics being developed that modulate Wnt signaling in cancer and discusses potential shortcomings associated with available therapeutics.

## 1. Introduction

Wnt signaling and its components have been implicated in a wide spectrum of important biological phenomena, where either a deficiency or overactivation of key effectors can lead to developmental disorders or impact cancer risk. The emergence of Wnt signaling, an integral mode of cell-to-cell communication, can be traced back to the early discovery of the first mammalian Wnt gene, *Int1*, discovered in 1982 [1]. Following the discovery of *Int1*, some of the most impactful discoveries were made which linked the Wnt pathway with developmental biology. Many of the genes regulated via the Wnt pathway, which were initially discovered to be important for development, later turned out to be oncogenes and tumor suppressors when studied in human cancer. Subsequently, some of the early genetic screens involving mutations in *armadillo* (β-catenin in vertebrates) and *dishevelled* (*dsh* in *Drosophila*) were similar to the *Wingless* mutants and thus shown to play an important role in segment polarity and embryonic development. Soon developmental assays, such as axis duplication assays in *Xenopus*, were shown to be excellent experimental model systems to characterize different components of the Wnt pathway. This led to classification of positive regulators of the Wnt pathway, for instance, wherein injection of murine Wnt1 mRNA into the embryo could induce axis duplication. Similarly, axis duplication was also induced by other components such as β-catenin, TCF, LEF, and GSK3β. In contrast, injection of Axin and APC mRNA resulted in complete loss of duplication, thereby identifying the negative regulators of the Wnt pathway. There was no major connection between the Wnt pathway and human cancer, until 1993 when a few research groups [2,3,4] reported the role of tumor suppressor APC and β-catenin in the Wnt pathway. Later, mutations in APC, β-catenin, and Axin were associated with genetic diseases such as familial adenomatous polyposis (FAP) in patients. These findings, for the first time, established a direct link between the Wnt pathway and human cancer.

Since the early discoveries, several genetic and biochemical studies have identified novel signaling components and provided deeper insights into the Wnt pathway. The known components of Wnt signaling include Wnt ligands, receptors and co-receptors (Frizzled and LRP), components of β-catenin destruction complex, and other transcriptional regulators within the nucleus. With recent advancements in structural biology and comprehensive genomic studies, it is clearly evident that the Wnt pathway is integral for developmental biology and plays a critical role in human cancer. In this review, we provide an overview of the key components of the Wnt signaling pathway and the milestones in both developmental and cancer biology. We also describe some recently discovered components and functions and emerging links of Wnt signaling with cancer stemness and antitumor immunity. Finally, we touch upon some strategies in development being tested in clinical trials to target the Wnt signaling pathway.

## 2. Overview of the Wnt Signaling Pathway

The Wnt pathway is one of the major signaling cascades that contributes to both normal development and pathophysiology. Herein, we will discuss the key components of the Wnt pathway and refer to some excellent reports which have laid the foundation and some recent findings that provide deeper insights into the complexities of this integral pathway.

The Wnt pathway is divided into two branches: β-catenin dependent (canonical) and independent (noncanonical), as outlined in Figure 1. The scope of Wnt signaling involvement in normal development and pathophysiology is daunting. With the 19 Wnt ligands, 10 Frizzled (FZD) receptors, and 3 Dishevelled (DVL) proteins participating in signaling, the amount of information relayed is enormous. Typically, in the canonical branch, Wnt proteins (secreted glycoproteins) bind to a seven-pass transmembrane receptor protein called FZD and low-density lipoprotein receptor-related protein 5/6 (LRP5/6) to activate the Wnt signaling pathway at the plasma membrane. In the absence of Wnts, β-catenin is degraded via a destruction complex consisting of tumor suppressors such as adenomatous polyposis coli (APC), Axin, glycogen synthase kinase-3β (GSK3β), and casein kinase 1 (CK1). This destruction complex induces phosphorylation of β-catenin on key serine and threonine residues (Ser33, Ser37, Ser 45, and Thr41) by CK1 and GSK3β, resulting in the ubiquitination by E3 ubiquitin ligase β-TrCP, and subsequent proteasomal degradation of β-catenin. Conversely, Wnt protein secretion results in activation of FZD and LRP5/6 receptors on the plasma membrane, permitting binding of DVL proteins. Sequential phosphorylation of cytoplasmic motifs on LRP5/6 receptors allows its interaction with Axin, resulting in destabilization of β-catenin destruction complex. Eventually, dephosphorylated β-catenin becomes stabilized and translocates into the nucleus to interact with transcription factor/lymphoid enhancer-binding factor (TCF/LEF) to initiate transcription of Wnt target genes. The β-catenin and TCF complex mediated transcriptional activation plays a crucial role in regulation of diverse cell behaviors, including cell fate, cell survival, proliferation, and stem cell renewal. In recent years, novel findings about the canonical Wnt pathway allowed the model to be refined and provided deeper insights into how this pathway is regulated. For instance, we now understand that proper production and secretion of Wnt ligands is a crucial step for Wnt pathway activation. An ER resident enzyme such as Porcupine, an acyl-transferase, is essential for the attachment of palmitoleic acid to Wnt ligands. This allows the lipid-modified Wnt ligands to bind to transmembrane protein Evi/WIs which are shuttled to plasma membrane for secretion [5]. Interestingly, a variety of mechanisms have been proposed for the short-range versus long-range release of Wnt ligands that may correspond to the role of Wnt in either development or cellular maintenance. For example, depending on the need, the levels and the range of Wnt signaling vary between the process of intestinal organoid and sperm maturation, suggesting that tissue-specific mechanisms may exist. Beyond the Wnt ligands, several other regulators have been demonstrated to play a crucial role in Wnt activation. For instance, R-spondin ligands serve as positive regulators and bind to leucine-rich repeat-containing G-protein-coupled receptor (Lgr4-6). In the absence of R-spondin, ZNRF3/RNF43, two homologues of E3 ubiquitin ligases, bind to FZD and target it for degradation. However, when R-spondin is present, it interacts with Lgr4-6, inhibiting the activity of ZNRF3/RNF43 and thereby allowing accumulation of FZD receptors on the plasma membrane. It is fascinating to see how Wnt target genes such as ZNRF3 and RNF43, function as negative feedback regulators in Lgr5-positive cells. Moreover, recent studies have highlighted the importance of R-spondin, Lgr5, and RNF43 in different cancers types, including colorectal cancer, which harbor inactivating mutations on RNF43 [6]. A surprising number of cytoplasmic Wnt regulators, including APC, Axin, and DVL, have been found to localize within the nucleus. APC and Axin, have been found to contain nuclear import and export sequences that direct them to shuttle in and out of the nucleus [7]. While the classical model asserts that Dishevelled functions in the cytosol, recent studies have shown that DVL proteins translocate into the nucleus via a regulatory post-translational acetylation switch in breast cancer cells [8,9]. Recent findings indicate that oncogenic Yes-associated protein (YAP) of the Hippo pathway paradoxically suppresses Wnt activity. The study reported that Wnt scaffolding protein Dishevelled (DVL) is responsible for cytosolic translocation of phosphorylated YAP [10]. This mechanistic study complements previous evidence in showing that YAP is a part of β-catenin destruction complex, thereby acting as a negative regulator of Wnt signaling. Recent findings have shown that alternative tyrosine-kinase related receptors (such as MET, FER, and FYN), and nonreceptor tyrosine kinases (such as SRC and ABL) could enhance of β-catenin and TCF complex mediated transcription by disrupting interaction of E-cadherin with β-catenin. Additionally, G-protein-coupled receptor signal transduction and environmental conditions such as hypoxia and high glucose levels could activate TCF-β-catenin signaling [11].

The noncanonical branch is independent of β-catenin and, even though less characterized, regulates more diverse cellular function such as cell organization and polarity, allowing it to be further classified into the planar cell polarity (PCP) pathway and Wnt/Ca^2+^ pathway. In the PCP pathway, Wnt ligands (usually Wnt 5a and Wnt 11) bind to panel of receptors including FZD and tyrosine kinase co-receptors such as ROR1/2 and Ryk. DVL relays noncanonical Wnt signals and interacts with Rac1 and DVL-associated activator of morphogenesis 1 (DAAM1). Rac1 activates downstream c-Jun kinases, while DAAM1 activates Rho to further activate Rho-associated kinase (ROCK), eventually regulating actin polymerization and cellular cytoskeletal arrangements. In the Wnt/Ca^2+^ pathway, the Wnt ligands bind to FZD which interacts with heterotrimeric G-proteins and DVL, resulting in activation of phospholipase C (PLC) and an intracellular increase in Ca^2+^ levels. This cascade results in downstream activation of downstream signaling proteins such as protein kinase C (PKC), calcineurin, and Ca^2+^/calmodulin-dependent protein kinase II (CaMKII) that regulate cell adhesion and cell migration. Wnt5a-mediated activation of the Wnt/Ca^2+^ branch antagonizes canonical Wnt/β-catenin signaling by phosphorylating TCF4 via Nemo-like kinase, thus preventing the binding of the β-catenin-TCF4 complex to DNA [12,13], suggesting a coordinated interplay between the different branches of the Wnt pathway.

Together, Wnt signals incorporated into the canonical and noncanonical pathways regulate complex normal cellular processes such as cell differentiation, development, tissue homeostasis, and wound healing; however, when the Wnt pathway is aberrantly regulated, it can be associated with developmental disorders, tumorigenesis, and other diseases.

## 3. Wnt Signaling in Human Diseases

Since Wnt signaling coordinates cell development processes and adult tissue homeostasis, it is certain that its deregulation can be linked with developmental disorders, cancer, and other diseases. Table 1 provides a list of some of the diseases associated with mutations in the Wnt signaling components. These include mutations in various Wnt ligands and components involved in both canonical and noncanonical branches of the Wnt pathway, shedding light on Wnt regulation in human development. For instance, increased expression of *Wnt1* results in synaptic rearrangement in patients with schizophrenia. Tetra-amelia, a rare human genetic disease characterized by absence of four limbs, has been proposed to be caused by a nonsense mutation in the *Wnt3* gene. Similarly, mutation in the *Wnt4* gene is linked to intersex phenotype, Mullerian-duct regression, and kidney developmental defects. Other development disorders such as Fuhrmann syndrome, Al-Awadi/Raas-Rothschild/Schinzel phocomelia syndrome, and Santos Syndrome are linked with loss-of-function mutation in *Wnt7A* gene. Mutations in *Wnt5B* and *Wnt10B* may be linked to type II diabetes and obesity, respectively. Aberrant Wnt signaling may play a role in precancerous conditions such as hypohidrotic ectodermal dysplasia and odonto-onycho-dermal dysplasia as a result of Wnt10A mutations. Mutations in either *Fzd4* or *LRP5* genes are associated with familial exudative vitreoretinopathy (FEVR). Moreover, heterozygous mutations in *Fzd2* are implicated to be involved with cardiovascular diseases and a rare skeletal disorder (known as omodysplasia) characterized by severe limb shortening and facial dysmorphism. The Wnt pathway is also involved in controlling bone mass, a discovery that stemmed from analysis of *LRP5* mutation in patients with osteoporosis-pseudoglioma syndrome (OPPG), a disorder characterized by low bone-mass density. Conversely, patients who harbor *LRP5* gain-of-function mutations experience high bone mass disease. Moreover, *LRP6* mutations are also linked with neuronal and metabolic disorders such as Alzheimer’s and coronary artery disease. The central mediators of the Wnt pathway, *Dishevelled* genes, are also implicated in human disease. Early discoveries suggested that *DVL* genes were imperative for segment polarity in wing hair of *Drosophila* and *Xenopus* embryos [14]. Moreover, *DVL* knockout mice were shown to exhibit abnormal social interaction in nest building, home cage huddling, neural tube closure, and cardiovascular malformations [15,16,17]. In humans, mutations in *DVL* are associated with severe disorders such as Schwartz-Jampel syndrome, Charcot-Marie-Tooth disease type 2A, DiGeorge syndrome, and Hirschsprung’s disease [18,19]. Recent promising evidence suggests that frameshift mutations on C-terminal tail of all three DVL genes can cause Robinow syndrome, characterized by skeletal abnormalities [20,21,22,23,24]. To summarize, DVL plays an important role in development, and mutations in DVL genes can lead to severe phenotypic defects. Recently, components of Wnt signaling such as DVL were reported to play an instrumental role in developing neural circuits and adult brain function and cause neurodevelopmental disorders such as autism spectrum disorders (ASDs) and intellectual disability (ID) [25].

Deregulation of Wnt/β-catenin signaling also predisposes patients to multiple cancer types such as colorectal, hepatocellular carcinoma, ovarian, and lung cancer. In particular, loss-of-function mutations in *APC, Axin1*, and *Axin*, have been well documented in the cancer types mentioned above. Conversely, gain-of-function mutation of β-catenin has been established in colorectal cancer with wild-type *APC*. In contrast, attenuated β-catenin signaling may be implicated in the development of Alzheimer’s disease, a neurodegenerative disease characterized by deposits of the amyloid β-peptide and selective death of neurons. Recently, increasing evidence has shown that *GSK-3β* may be a key link between diabetes mellitus (DM) and Alzheimer’s disease (AD), where GSK-3β controls glycogen synthesis, thereby regulating blood glucose [26]. Genome-wide association studies have illustrated the role of *TCF4* in these seemingly diverse disorders. *TCF4* gene has been strongly implicated in type II diabetes; however, recent data suggest that *TCF4* is also an important regulator of neurodevelopment disorders such as schizophrenia, Fuchs’ endothelial corneal dystrophy, and primary sclerosing cholangitis. However, rare *TCF4* mutations causing Pitt–Hopkins syndrome, a disorder characterized by intellectual disability and developmental delay, have also been described in patients with other neurodevelopmental disorders [27].

Interestingly, since components of noncanonical branch also play a central role in the cell processes and stress response, mutations in JNK, Rho/Rac, TIAM1, and NFAT have been associated with various disorders, including metabolic, neurodegenerative, and cardiovascular diseases. JNK is one of the most investigated signal transducers, and emerging evidence suggests that different isoforms of JNK (JNK1 and JNK2) may promote the development of obesity to insulin resistance in a cell-specific manner, NAFLD, and type II diabetes. This has led to development of isoform-specific JNK inhibitors with specific tissue distribution as possible drug targets for the treatment of type II diabetes [28]. Moreover, recent evidence suggests that excessive activity of the RhoA/Rac-kinase pathway, widely known to play important roles in many cellular functions, promotes the development of AD pathogenesis and cardiovascular diseases [29,30]. Recently, mutations in the hematopoietic specific GTPase, *RAC2*, have been found to cause a human disease, a severe phagocytic immunodeficiency characterized by life-threatening infections in infancy. Interestingly, the phenotype was predicted by a mouse knockout of Rac2 and resembles leukocyte adhesion deficiency (LAD) [31]. The activity of T-cell lymphoma invasion and metastasis 1 (TIAM1), a Rac guanine nucleotide exchange factor (GEF), crucial for cell adhesion and migration, has been demonstrated to be upregulated in some cancers [32,33]. Additionally, *TIAM1* has also shown to play a pivotal role in cardiac hypertrophy associated with heart failure [34]. Mathematical models and in vivo studies predict that restriction of nuclear occupancy of NFAT transcription factors, resulting in inactivation of NFAT target genes, may cause Down’s syndrome (chromosomal trisomy), further causing neurological, skeletal, cardiovascular, and immunological defects [35]. More generally, these findings suggest that the destabilization of important regulatory pathways such as Wnt signaling can underlie human diseases.

## 4. Wnt Signaling in Cancer

### 4.1. Hematological Malignancies

Blood or hematological malignancies such as leukemia, lymphoma, and myeloma originate from uncontrolled expansion of hematopoietic stem cells (HSCs) which exhibit deregulation of the Wnt signaling pathway [86,87]. In many instances, the critical role of Wnt signaling in leukemia stem cells has been elucidated and accounts for chemotherapeutic drug resistance and relapse. Wnt signaling activates self-renewal and proliferation of hematopoietic and progenitor cells, suggesting its involvement in hematological malignancies.

Leukemia is frequently associated with mutations of receptor tyrosine kinases (RTKs) and chromosomal translocations, where both mutations have been associated with aberrant Wnt signaling. Expression of WNT3 and LEF1 is amplified in chronic lymphocytic leukemia (CLL) B-cells, a subset of leukemias, compared to normal B-cells [88,89]. Other Wnts such as *Wnt4*, *Wnt5B*, *Wnt6*, *Wnt7B*, *Wnt9A*, *Wnt10A*, *Wnt14*, and *Wnt16* are robustly expressed in CLL B-cells and acute lymphoblastic leukemia (ALL) cells, whereas these can be scarcely detected in normal pre-B cells [90,91,92]. Similarly, high levels of receptor Ror1 are associated with CLL when compared to nonleukemic leucocytes. Some of the negative regulators of Wnt signaling, such as the *Dickkopf* (*Dkk*) and secreted Frizzled receptor protein (SRFP) families, are reported to be silenced by promoter hypermethylation, thereby demonstrating a novel mechanism of Wnt activation in t(8;21)-leukemia cells [93,94,95,96,97]. Moreover, different subsets of AML, ALL, and *BCR-ABL* chronic myeloid leukemia (CML) cells rely on both β-catenin-mediated signaling and Wnt-mediated calcium signaling for self-renewal, proliferation, and survival [98,99]. Recently, a study by Saenz et al. demonstrated an increase in nuclear β-catenin and TCF4 transcriptional activity in post-myeloproliferative neoplasms secondary to AML. Moreover, knockdown of β-catenin in AML induced apoptosis of both JAKi-sensitive and JAKi-resistant blast progenitor cells. Preclinical in vitro and in vivo findings highlight a novel therapeutic approach of co-targeting β-catenin and BETP antagonists (bromodomain and extraterminal domain protein) for AML treatment [100].

Blood cancers arising from lymph nodes are collectively termed lymphoma, including Hodgkin’s lymphoma, mantle cell lymphoma (MCL), and Burkitt’s lymphoma. Several Wnt components such as *cyclinD1*, *TCF7*, *FZD7*, *LRP5*, *AXIN1*, *APC*, and *DVL3* were upregulated in MCL cells. Moreover, MCL cells show inactivation of GSK3β resulting in increased active β-catenin levels in the nucleus [101,102]. Patients with lymphoma show overexpression and nuclear accentuation of β-catenin in 46.67% of the diffuse large B-cell lymphoma (DLBCL) cases [103,104,105]. Evidence suggests that upregulation of transcription factors TCF1 and LEF1 in T-cell and small B-cell lymphoma may confer chemoresistance, suggesting a possibility of reliance on the Wnt pathway for cancer stem cells in lymphoma [106,107]. Furthermore, absence of Wnt5a can lead to lymphoma by activating β-catenin signaling [108,109]. However, more studies are needed to define the dependence of lymphoma progression on different Wnt-mediated cues to define the potential of developing therapeutic interventions for patients.

Myeloma is cancer of the plasma cells, and similar to leukemia and lymphoma, Wnt activation is observed in myeloma cells. It has been established that several Wnt ligands are overexpressed in multiple myeloma (MM) cell lines [110]. In addition, Wnt 3a ligands are able to induce both β-catenin signaling and RhoA signaling, which allows cytoskeletal rearrangement and planar cell polarity of MM cells [111]. This suggests that MM cells rely on both β-catenin-dependent and β-catenin-independent Wnt signaling for proliferation and survival. Taken together, these studies exemplify that Wnt signaling is activated in different hematological malignancies leading to enhanced proliferation and self-renewal capacities of cells.

### 4.2. Breast Cancer

One of the early links between Wnt signaling and breast cancer was made using genetically modified mouse models, where mouse mammary tumor virus (MMTV) was integrated into loci of Wnt genes [112]. Some of the proto-oncogenes that were frequently activated by MMTV insertion included *Wnt1*, although *Wnt 3* and *Wnt 10B* were also found to be activated in similar manner [113,114]. Unlike other human cancers, mutations on Wnt components such as *APC*, *Axin*, and *CTNNB1* (encoding β-catenin) are very rare. However, dysregulated canonical and noncanonical Wnt signaling branches have been implicated in various breast cancer subtypes. It is considered that alteration of Wnt signaling in breast tumorigenesis occurs via epigenetic changes, wherein the positive components of the Wnt pathway are increased and the antagonists (negative regulators) are decreased at the levels of ligands, receptors, and transducers [13,115]. Few Wnt ligands such as Wnt5b have been identified as key regulatory factors for activating both canonical and noncanonical signaling components in basal-like breast cancer (BLBC). This study identifies Wnt5b as a key regulator for activating Wnt target genes and EMT markers such as MET, CD44, TCF4, and TWIST1/2, to name a few. Wnt5b knockdown studies conducted via shRNA or by using Porcupine inhibitors resulted in inhibition of tumorigenicity in xenograft model, further signifying the critical role of Wnt5b in BLBC [43]. Some of the evidence implicating Wnt signaling in breast cancer includes elevated levels of β-catenin in the nuclear or cytoplasmic compartments of cancer cells. Several reports have shown elevated levels of β-catenin in total cell lysate as well as in the nuclear compartment, compared to the normal tissues. It was further suggested that β-catenin levels are correlated with elevated expression of cyclinD1 and c-myc, downstream Wnt target genes [116,117,118]. Collectively, these studies suggest that elevated β-catenin could serve as a prognostic marker in breast cancer by increasing transcriptional expression of tumor-promoting genes, which could contribute to the ability of dysregulated Wnt signaling in breast cancer development. Recent reports focusing on DVL proteins suggest that their nuclear localization is important in tumor development. Another study provided evidence that Wnt5a represses ribosomal RNA (rRNA) synthesis by promoting DVL-1 accumulation in nucleolus of breast cancer cells, where nucleolar DVL-1 releases Pol I transcription activator and SIRT7 from rDNA loci, thereby inhibiting tumor growth [119]. Moreover, SIRT1 loss of function results in a decrease in DVL protein levels, causing an inhibition in downstream Wnt target gene expression and Wnt-stimulated cell migration in several cancer models [120]. Additionally, novel insights recently reported by our group demonstrate that a DVL-1 post-translational acetylation promotes nuclear localization in triple-negative breast cancer, highlighting a novel molecular switch that regulates subcellular localization of DVL proteins [9]. Further investigation demonstrates a differential role of nuclear DVL1 and DVL3 in regulating tissue-specific aromatase promoters [8] and DVL1 binding to *FZD7* promoters in breast cancer models [121]. LEF1, a downstream mediator of the canonical Wnt pathway, is also considered a promising target due to its involvement in stem-cell maintenance and promotion of EMT phenotype in different cancer types [122]. Moreover, the Wnt pathway is becoming popular for its implication in cancer stem cell maintenance [123]. Mechanistic studies show that certain *Wnt 3A* genes are highly enriched in breast cancer stem cell population in comparison to normal-breast stem cell population, resulting in high metastatic potential and recurrence rate in breast cancer patients [124]. A clinical study conducted using breast cancer (BC) tissue from 44 women and tissue from 25 healthy women suggested that the number of stem cells (CD44 high and CD24 low) was significantly associated with high expression levels of Wnts and β-catenin, thereby indicating a potential for targeting Wnt signaling for treatment of breast cancer [125].

### 4.3. Colorectal Cancer

The Wnt pathway is one of the key signaling cascades responsible for carcinogenesis in colorectal tissues. It is widely accepted that almost 80% of the colorectal cancer (CRC) patients harbor inactivating mutations in the *APC* gene along with β-catenin mutations that result in hyperactivation of the Wnt signaling pathway [126]. The progression of CRC from normal colonic epithelium occurs via several genetic and epigenetic mutations that allow propagation of Wnt target genes involved in cell migration, invasion, and proliferation and stem cell renewal. Loss-of-function mutations of *APC* (most being frameshift and nonsense mutations) result in a protein truncation which is believed to hyperactivate Wnt signaling by disrupting the destruction complex mediating β-catenin degradation. Other negative regulators (or tumor suppressors) such as *Axin1* and *Axin2* often undergo mutations and lead to genetic disposition to CRC development. Negative regulators, such as RNF43 and ZNRF3, promote FZD receptor turnover resulting in proliferation of Paneth cells by activating the Wnt cascade [127,128]. Activating mutations on β-catenin protein (at serine and threonine residues preventing subsequent ubiquitination) and high levels of β-catenin in the nucleus also promote CRC progression in approximately 1% of the cases [129,130]. Furthermore, elevated levels of β-catenin in the nucleus are positively correlated with target oncogenes such as c-myc and cyclinD1, which play a very important role in cancer progression in CRC patients. Other Wnt target genes that regulate EMT such as MMP7 are expressed in about 90% of the CRCs resulting in unfavorable outcomes, higher metastatic potential, and chemoresistance [131]. Other members of the Wnt pathway, belonging to TCF and LEF families, have been reported to undergo mutations in CRC; however, they have not been fully characterized [132].

Furthermore, recent studies have elucidated the importance of oncogenic Wnt signals and their effect on metastasis and chemoresistance in CRC. Recently, a study reported that Wnt2B co-operates with FZD7 receptor to initiate mesenchymal-to-epithelial transition (MET), a reversible transition at a secondary site, using an LIM 1863-Mph three-dimensional in vitro model [133]. The role of Wnt5a represents another example of how Wnt pathway stimulation, depending on the branch of signaling engaged and the cellular context, may not always be associated with promoting cancer progression. Wnt5a is involved in mediating noncanonical signaling via FZD or ROR receptors and has been associated with both oncogenic and tumor-suppressive effects, as discussed in other reviews [134]. Moreover, TCF7L2, a Wnt transcription factor, is overexpressed in primary rectal cancer and has been shown to impart resistance to chemoradiotherapy (CRT). The study reported that inhibition of β-catenin by siRNA or small-molecule inhibitor (XAV-939) resulted in sensitization of RPE-1 cells to CRT. Conversely, cells expressing degradation-resistant mutant of β-catenin (S33Y) also boosted resistance of RPE-1 cells to CRT [135]. Gene enrichment analysis performed using The Cancer Genome Atlas (TCGA) shows a close relation between the Wnt pathway and T-cell signature genes. Recently, it was shown that β-catenin signaling is inversely associated with T-cell infiltration in colorectal tissues. This study analyzed 155 colorectal cancer tissues, where tumors with high β-catenin levels had significantly less CD8+ T-cell infiltration. The study also showed regulation of CCL4 expression by β-catenin impaired the recruitment of CD103+ dendritic cells for CD8+ T-cell activation. These findings may provide another mechanistic clue behind resistance to immunotherapy in colorectal cancer patients [136]. MicroRNAs (miRNAs) have been linked with Wnt signaling and modulation of tumorigenesis in multiple cancers including colorectal cancer. For instance, β-catenin is positively associated with miR-19a expression, which attenuates inhibitory effects of APC on cellular migration and invasion in CRC patients. Moreover, overexpression of miR135b induces *APC* loss and is associated with poor clinical outcomes in human CRC cells. For more information about miRNA and its association with Wnt signaling in the pathogenesis of CRC, we refer the readers to a review by Rahmani et al. [137].

## 5. Insights into the Interplay between Wnt Signaling, Antitumor Immunity, and Stemness

### 5.1. Wnt Signaling and Antitumor Immunity

Wnt signaling is an important regulator of proliferation and differentiation of T-cell development and peripheral immune responses. Since this section will briefly overview the role of the Wnt pathway in cancer immunomodulation, we point the readers to previous reviews that have highlighted the role of Wnt signaling in the regulation of immune cells [138,139]. Dysregulation of the Wnt pathway has been implicated in actively suppressing antitumor immunity in several tumor types by modulating the tumor microenvironment and immune cell infiltration. For example, Dkk1 derived from stromal cells within the tumor has been shown to downregulate β-catenin in myeloid-derived cells, leading to a suppressed T-cell response and tumor growth [140]. A Dkk1 vaccination study in a murine model of myeloma showed higher CD4+ and CD8+ T-cell immune activity and provided a strong rationale for using Dkk1 as a myeloma immunotherapeutic target [141]. Another study using a conditional knockout mouse model for *Wnt5a* and its receptor *Ror2* reports attenuation of inflammatory cytokine production and a decrease in interferon-γ (IFNγ)-producing CD4+ Th1 cells in the colon. Likewise, in vitro experiments demonstrated that the Wnt5a and Ror2 signaling axis promotes IFNγ production, augmenting IL-12 expression in dendritic cells and thereby inducing Th1 differentiation in colitis [142]. Moreover, cytoplasmic regulators such as DVL1 and DVL3 have also been associated with immune modulation. A study using *Dishevelled 1* knockout mice displayed reduction and mislocalization of Paneth cells and an increase in CD8+ T cells in the lamina propria, resulting in abnormal gut microbiota [143]. Moreover, inhibition of Wnt3a or DVL3 significantly increases production of proinflammatory cytokines (IL-12, IL-6, and TNFα) and reduces β-catenin accumulation, suggesting that the Wnt3a–DVL3–β-catenin signaling axis could be a potential intervention target for manipulating inflammatory responses [144]. Immunohistochemical analysis of colorectal cancer tissues showed that tumors with high β-catenin expression showed a significant reduction of CD8+ T-cell infiltration. Furthermore, the authors suggested that β-catenin could regulate CCL4 expression to recruit CD103+ dendritic cells, thereby influencing CD8+ T-cell activation [136]. In another set of studies, overexpression of active β-catenin in melanoma or dendritic cells resulted in secretion of anti-inflammatory cytokine (IL-10), impairing the ability of dendritic cells to activate CD8+ cytotoxic T lymphocyte cells for tumor recognition [145,146]. Moreover, gene set enrichment and TCGA analyses conducted across multiple cancers show an inverse correlation between gene expression of Wnt/β-catenin components and T-cell signature genes, indicating that Wnt signaling allows tumor immune evasion by suppressing immune cell activation, infiltration, and function in the tumor microenvironment [147]. Collectively, these studies suggest that oncogenic Wnt signaling may favor tumor progression by modulating antitumor immunity and suggest a benefit of combination therapy targeting the Wnt pathway and immune checkpoint inhibitors.

### 5.2. Wnt Signaling in Cancer Stem Cell Biology

Cancer stem cells (CSCs) are endowed with self-renewal capacity that results in tumor initiation, drug resistance, metastasis, and cancer relapse, eventually leading to cancer-related deaths. Some of the early reports linking cancer stem cells were in breast and colorectal cancer models [148], a main focus of this section of the review. High Wnt activity has been linked with a CSC phenotype in different tumor types using canonical Wnt reporter activity and gene expression studies [149]. For instance, the expression of *CD44*, a Wnt target gene, can be detected in primary and metastatic malignancies, where tumor cells with elevated levels of CD44 have been shown to behave as CSCs. Canonical Wnt3a-mediated signaling has been shown to support stem-cell-like properties and sphere-forming ability of breast tumor cells [124]. Similarly, *Lgr5* is another example of a Wnt target gene that is prominently discussed in the CSC context. Lgr5 is a cell surface G-protein-coupled receptor which binds R-spondin to augment Wnt-ligand-mediated Wnt signaling. Evidence indicates the R-spondin and Lgr5 complex inhibits the E3 ubiquitin ligases, namely Rnf43 and Znrf3, and thus enhances Wnt signaling, thereby likely promoting stem cell properties. Interestingly, overexpression or inhibition of Lgr5 results in increased or decreased stem cell functionality in breast cancer, respectively [150]. In colorectal cancer, R-spondin 2 suppresses Wnt signaling in an Lgr5-mediated manner, resulting in a reduction in stem cell properties [151]. Together, these findings suggest that the functionality and the use of R-spondin and Lrg5 as a CSC marker can be cancer-context-dependent. Other intrinsic regulators of the Wnt pathway such as proliferative cell nuclear antigen associated factor (PAF), *AXIN2*, and *DKK1* have shown to enhance the CSC pool, potentially by activating Wnt activity [152,153,154]. In a similar fashion, microRNA (miR) and long noncoding RNA (lncRNA) have also been implicated in CSC by modulating the Wnt signaling pathway. Some examples include miR-21, miR-451, miR-142, and lncTCF7, which have been shown to upregulate stem-cell-like properties by modulating the Wnt pathway regulators such as cyclinD1/c-myc levels, cyclooxygenase-2 (COX-2), and β-catenin and by recruiting SWI/SNF complex to TCF promoter, respectively [155,156,157,158,159]. Moreover, crosstalk of the Wnt pathway with TGFβ signaling has been shown to further induce change in stem-cell-like phenotype by activating epithelial–mesenchymal transition in breast cancer cells [160]. Collectively, these findings indicate the importance of Wnt signaling in cancer stem biology, highlighting some of the Wnt-mediated molecular mechanisms for promoting CSC function.

## 6. Wnt Pathway Inhibitors and Efforts in Clinical Trials

Aberrant regulation of the Wnt pathway has been linked with worse cancer outcomes, and therefore, current strategies aim at targeting this signaling cascade in multiple tumor subtypes [11]. Pharmacological inhibitors of the Wnt pathway can be broadly classified into three groups: inhibitors of Wnt-receptor complex, regulators of β-catenin destruction complex, and inhibitors of TCF/β-catenin transcription complex. Herein, we provide a simplistic overview of the small-molecule inhibitors and antibodies designed to modulate the Wnt pathway in Figure 2 and some of the ongoing clinical trials in Table 2. We encourage the readers to read other research articles for detailed descriptions of the efficacy and safety outcomes [11,161,162] as well as the structure of Wnt pathway inhibitors [163,164,165,166].

### 6.1. Inhibitors of Wnt-Receptors Complexes

The Wnt-pathway-associated transmembrane proteins and ligands are targeted by monoclonal antibodies (mAbs), antibody-drug conjugates (ADCs), decoy receptors, and small-molecule inhibitors. Currently, only a few of these agents have reached the clinical stages of development. Monoclonal antibodies developed against Wnt1 ligand have been shown to induce apoptosis and reduction in downstream protein components, leading to tumor suppression in human cancer cells [167,168].

Vantictumab (OMP-18R5), a first-in-class antibody, targets the extracellular domain of five Fzd receptors (Fzd1, Fzd2, Fzd5, Fzd7, and Fzd8), thereby suppressing the induction of canonical Wnt signaling. This agent exhibited reduction in cancer stem cell population and decreased tumor growth in PDX models of various tumor types, both alone and in combination with standard-of-care chemotherapeutic agents [169]. Vantictumab has been tested in clinical trials in patients with neuroendocrine tumors, metastatic HER2-negative breast cancer [170], and untreated metastatic pancreatic cancer [171]. Vantictumab was well tolerated in phase I clinical trials in patients with solid tumors; however, bone toxicities including pathological fracture occurrence were reported, and thus these trials were ultimately terminated due to concerns around bone-related safety [172].

Ipafricept (OMP-54F28) is a first-in-class recombinant fusion protein which comprises of extracellular domains of human Fzd8 fused to a human IgG1 Fc fragment, designed to block Wnt signaling through binding of Wnt ligands [173]. Xenograft studies in ovarian cancer show that ipafricept decreases the frequency of stem cells and suppresses tumor formation. Clinical studies indicate that pretreatment with ipafricept followed by administration of taxane-based chemotherapeutic agents (carboplatin and paclitaxel) results in 35% complete response, 18% stable disease, and 47% partial response [174,175].

Foxy-5 is a Wnt5A-mimicking peptide that activates the Wnt5A receptors Fzd2 and Fzd5 to modulate downstream Wnt5A signaling. In thyroid, prostate, colon, and breast cancer, Wnt5A is suggested to have a tumor-suppressor role wherein reconstitution of Wnt5A impairs tumor metastasis by targeting cell motility and, therefore, is associated with a higher disease-free survival [176]. By contrast, in gastric cancer, high Wnt5A expression is associated with poor prognosis and increased metastasis and invasion [177]. Based on these findings, an ongoing phase 1 study is being conducted to evaluate the safety and tolerability of treatment with Foxy-5 in patients with metastatic breast, colorectal, or prostate cancer, and only Wnt5a-negative or Wnt5a-low patients are enrolled in the study [178].

Porcupine inhibitors are selective inhibitors such as WNT974 and inhibitors of Wnt production (IWPs) that have been developed to disrupt Wnt secretion and tumor growth. Wnt proteins are post-translationally palmitoylated by a membrane-bound O-acyltransferase (MBOAT) called Porcupine, which is critical for Wnt ligand secretion. Porcupine inhibitors block tumor growth by retention of immature, nonpalmitoylated Wnt ligands in the endoplasmic reticulum. Preclinical studies have shown promising results with Porcupine inhibitors such as WNT974 (also known as LGK974) that antagonize Wnt signaling and enhance targeting of chronic myeloid leukemia (CML) stem cell population. The efficacy of WNT974 was further enhanced in combination with tyrosine kinase inhibitor (nilotinib) and significantly inhibited the proliferation and growth of CML stem and progenitor cells in vivo, while sparing the normal counterparts [179]. WNT974 has also been tested in other combination therapies for metastatic, colorectal, and head and neck cancer with characteristic *RNF43/ZNRF3* mutations [180]. ETC-159 is another small-molecule Porcupine inhibitor in a phase I clinical trial that has been ongoing since 2015 [167].

### 6.2. Regulators of β-Catenin Destruction Complex

This class of inhibitors includes small-molecule antagonists that target components of the β-catenin destruction complex, including *DKK* and Dishevelled proteins.

Inhibitors of Wnt response (IWRs), such as IWR-1, and XAV939 induce protein stability of Axin by inhibiting the poly(ADP)-ribosylating enzymes tankyrase 1 and tankyrase 2 [181,182]. Mouse tumor xenografts with triple-negative breast cancer were incubated with paclitaxel-combined XAV939 regimen which induced cell apoptosis, resulting in inhibition of the Wnt signaling pathway and suppression of EMT and angiogenesis [183]. Similar studies performed in lung adenocarcinoma A549 cells and small-cell lung cancer suggest that XAV939 attenuates Wnt signaling by reducing c-myc and β-catenin levels, resulting in cell apoptosis [184,185]. Moreover, tankyrase inhibitors further increased apoptosis in combination with 5-fluorouracil/cisplatin, AKT/PI3K inhibitors, by alternation of β-catenin levels, Axin, and CSC markers, overcoming multidrug resistance in colon cancer cells [186,187]. Current phase II clinical trials are being conducted using 2X-121, a small tankyrase inhibitor, in metastatic breast cancer patients, and investigators are still awaiting the completion of this study.

Dishevelled inhibitors target the PDZ domain of DVL proteins which interacts with the carboxyl terminal end of the FZD receptors. Some compounds, such as NSC668036, FJ9, and 3289-8625, identified through in silico screening and nuclear magnetic resonance (NMR) have shown the ability to block Wnt signaling in the micromolar range in vitro [188,189,190].

### 6.3. Inhibitors of TCF/β-Catenin Transcription Complex

PRI-724 is an antagonist that interrupts interaction between β-catenin and transcriptional co-activator CREB-binding protein (CBP). Preclinical studies of PRI-724 in patients with advanced pancreatic cancer suggest this agent can decrease metastatic potential but has only modest clinical outcomes [191]. E7386 is a first-in-class orally active β-catenin/CBP modulator that potentially inhibits Wnt/β-catenin signaling in human HCC xenograft [192].

PFK115-584, CGP049090, and PKF222-815 are compounds identified by high-throughput ELISA that can perturb β-catenin interaction with both TCF and APC proteins [193]. Another high-throughput screen conducted in *D. melanogaster* identified three potential compounds, namely iCRT3, iCRT5, and iCRT14, as being able to disrupt β-catenin-TCF interaction. These Wnt inhibitors downregulate the expression of Wnt target genes but have been specifically shown to be cytotoxic to human colorectal cancer cells [194].

## 7. Concluding Remarks

Significant advances have been made, enabling a better understanding of canonical and noncanonical signaling networks of the Wnt pathway. We now understand the dynamic range of consequences of aberrant Wnt signaling that impact oncogenesis and developmental biology. From the literature cited and discussed at length here, it is clear that small-molecule or antibody-based drugs that regulate various branches of the Wnt pathway will likely prove beneficial in the treatment of a wide range of diseases. Combination therapies involving Wnt inhibitors and immunotherapy will likely represent an area ripe for further future investigation which may provide yet another means for treating a wide range of disorders, including autoimmune disease, chronic inflammation, and cancer. Moreover, targeting the Wnt pathway might prove effective in a preventive setting for individuals who have not yet acquired alterations that lead to advanced stages of disease. Another area for further investigation involves understanding the functional consequences of DVL post-translational regulation, which remains poorly studied; however, there is enough evidence to conclude that this area of investigation will likely lead to major insights into understanding how the various branches of Wnt signaling are engaged. Finally, it will be important to further investigate the role of Wnt signaling in innate and adaptive immunity associated with a wide range of pathophysiology. Several exciting emerging links of Wnt biology have been discussed with regard to cancer stemness, metastasis, and antitumor immunity, and future investigation of how Wnt signaling impacts these areas will be very important.

## Figures and Tables

**Figure 1 ijms-21-08018-f001:**
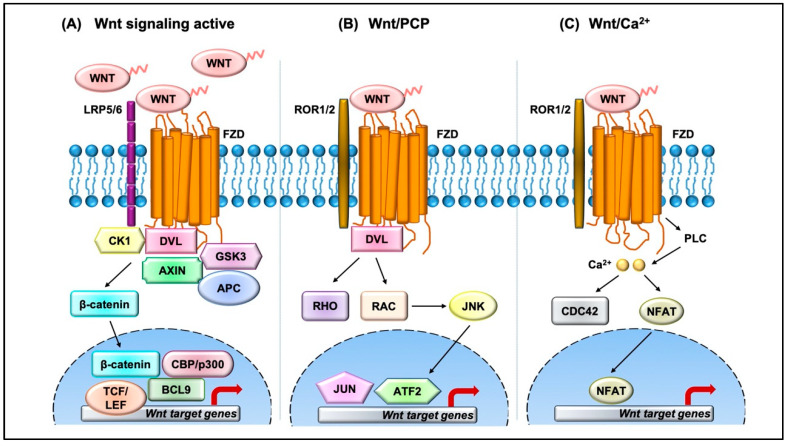
Overview of the Wnt signaling pathway. (**A**) In the canonical Wnt pathway, secreted Wnt ligands (usually Wnt3A and Wnt1) bind to Frizzled (FZD) receptors and LRP co-receptors. These receptors are then activated via CK1 and GSK3B mediated phosphorylation, which further recruits Dishevelled (DVL) to the plasma membrane and initiates activation. The DVL signalosome results in sequestration and inhibition of β-catenin destruction complex (Axin and APC) allowing stabilized β-catenin levels in the cytoplasm to increase. β-Catenin translocates into the nucleus where it forms an active complex with lymphoid enhancer factor (LEF), T-cell factor (TCF), and other histone-modifying co-activators such as CBP/p300 and BCL9, resulting in transcriptional activation (as represented by red arrow) of Wnt target genes which in turn causes an increase in cellular processes such as cell proliferation and differentiation and stem cell renewal. (**B**) In the Wnt/PCP pathway, Wnt ligands bind to FZD and co-receptors such as ROR1/2 and recruit DVL to the plasma membrane. DVL interacts with small GTPases such as RHO and RAC to further trigger activation of JNK. This results in activation of cytoskeletal rearrangements by transcriptional responses via JUN and ATF2 (activating transcription factor). (**C**) The Wnt/Ca^2+^ pathway is initiated by Wnt-FZD-ROR complex and G-protein-triggered phospholipase C (PLC) activity that results in intracellular Ca^2+^ influx. This further activates CDC42 and triggers calcium-dependent cell movement and polarity via various transcriptional responses (red arrow).

**Figure 2 ijms-21-08018-f002:**
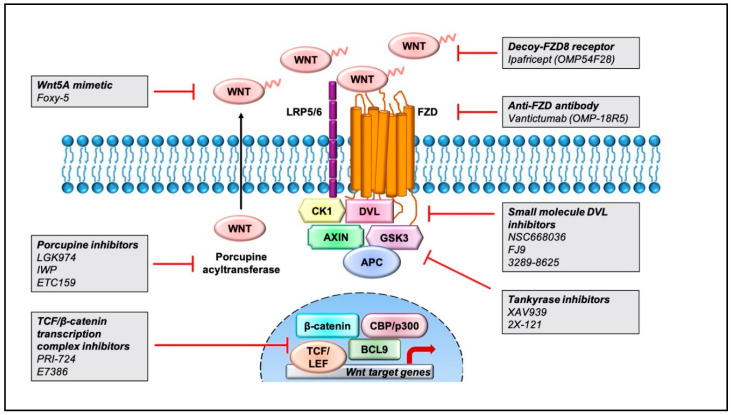
Pharmacological inhibitors targeting the Wnt pathway that are currently being tested for cancer treatment. Representation of pharmaceutical compounds that disrupt the Wnt pathway at several stages, including the Wnt-receptor complex, β-catenin destruction complex, and TCF/β-catenin transcriptional complex.

**Table 1 ijms-21-08018-t001:** Human diseases associated with dysregulation in Wnt signaling components.

Gene	Dysfunction	Associated Disease	References
*Wnt 1*	Gain of function	Schizophrenia	[36]
*Wnt 3*	Loss of function	Tetra-amelia	[37]
*Wnt 4*	Gain and loss of function	Intersex phenotype (GOF), kidney development, Mullerian duct regression and virilization (LOF)	[38,39,40,41]
*Wnt 5B*	Gain of function	Type II diabetes, breast tumorigenesis	[42,43]
*Wnt 7A*	Loss of function	Fuhrmann syndrome, Al-Awadi/Raas-Rothschild/Schinzel phocomelia syndrome, Santos syndrome	[44,45]
*Wnt 10A*	Loss of function	Hypohidrotic ectodermal dysplasia, odonto-onycho-dermal dysplasia	[46,47]
*Wnt 10B*	Loss of function	Obesity, reduced bone mass	[48,49]
*Fzd 2*	Heterozygous mutations	Omodysplasia, cardiovascular disease	[16,50]
*Fzd 4*	Loss of function	Familial exudative vitreoretinopathy (FEVR)	[51,52]
*LRP5*	Gain and loss of function	High bone mass (GOF), osteoporosis-pseudoglioma syndrome (LOF), familial exudative vitreoretinopathy (FEVR)	[53,54,55,56,57]
*LRP6*	Gain and loss of function	Alzheimer’s disease (LOF), coronary artery disease (LOF), osteoarthritis (GOF)	[58,59,60,61]
*DVL1*	Gain and loss of function	Robinow syndrome (frameshift mutation), Schwartz–Jampel syndrome, Charcot–Marie–Tooth disease type 2A and DiGeorge syndrome, myocardial infarction, Hirschsprung’s disease (GOF), autism spectrum disorders	[18,19,21,22,25]
*DVL2*	Loss of function	Robinow syndrome, defect in cardiac outflow tract formation	[17,23]
*DVL3*	Gain and loss of function	Hirschsprung’s disease (GOF), autism spectrum disorders	[19,25]
*Axin 1*	Loss of function	Caudal duplication anomalies, gastrointestinal cancers, colorectal cancer, hepatocellular carcinomas, sporadic medulloblastoma, breast cancer	[62,63,64,65,66,67,68]
*Axin 2*	Loss of function	Congenital heart defects, familial tooth agenesis, predisposition to multiple cancers including hepatocellular carcinoma and colorectal, prostate, ovarian, and lung cancers	[69,70,71,72,73]
*APC*	Loss of function	Familial adenomatous polyposis (FAP), colon cancer	[3,74]
*GSK3β*	Altered activity	Alzheimer’s disease, diabetes, schizophrenia, bipolar disorder, and cancer	[26,75,76,77,78,79]
*β-Catenin*	Gain and loss of function	Cancer (GOF), Alzheimer’s disease (LOF)	[80,81,82]
*TCF4*	Transcript variants	Pitt–Hopkins syndrome, schizophrenia, Fuchs’ endothelial corneal dystrophy, primary sclerosing cholangitis, type II diabetes	[27,83,84]
*JNK*	Altered activity	Obesity, type II diabetes, nonalcoholic fatty liver disease (NAFLD)	[28]
*Rho/Rac*	Altered activity	Alzheimer’s disease, cardiovascular disease, leukocyte adhesion deficiency (LAD)	[29,30,31]
*TIAM1*	Altered activity	Cardiovascular disease and cancer	[32,33,85]
*NFAT*	Loss of function	Down’s syndrome	[35]

**Table 2 ijms-21-08018-t002:** Overview of clinical trials for investigated agents targeting Wnt signaling pathway.

Agent	Target Component	Trial Phase and Identifier	Condition or Disease	Collaborator and Sponsor
**WNT974**	Porcupine inhibitor	Phase 1 NCT01351103	Pancreatic cancer, BRAF mutant colorectal cancer, melanoma, triple-negative breast cancer, head and neck squamous cell cancer, cervical squamous cell cancer, esophageal squamous cell cancer, and lung squamous cell cancer	Novartis Pharmaceuticals
**WNT974** (in combination with LGX818 and cetuximab)	Porcupine inhibitor	Phase 1 and 2 NCT02278133	Metastatic colorectal cancer with BRAFV600-mutant mCRC with RNF43 mutations or RSPO fusions.	Array BioPharma
**Foxy-5**	Wnt-5A mimetic	Phase 1 and 2 NCT02020291 NCT02655952 NCT03883802	Metastatic breast cancer, colorectal cancer, prostate cancer, colon cancer	WntResearch AB, SMS-Oncology BV, SAGA Diagnostics AB, Unilabs A/S, BioVica AB, Catalan Institute of Oncology
**ETC-159**	Porcupine inhibitor	Phase 1 NCT02521844	Solid tumors	EDDC (Experimental Drug Development Centre), A*STAR Research Entities, PPD
**Ipafricept** (OMP54F28)	Decoy Fzd8-receptor	Phase 1 NCT01608867	Solid tumors	Bayer; OncoMed Pharmaceuticals, Inc.
**Vantictumab** (OMP-18R5)	Anti-FZD antibody	Phase 1 NCT01345201 NCT02005315 NCT01957007 NCT01973309	Solid tumors, pancreatic cancer stage IV, pancreatic cancer, non-small-cell lung cancer, metastatic breast cancer	OncoMed Pharmaceuticals, Inc.
**2X-121**	Tankyrase inhibitors	Phase 2 NCT03562832 NCT03878849	Metastatic breast cancer, advanced ovarian cancer	Oncology Venture, Smerud Medical Research International AS, Danish Breast Cancer Cooperative Group, Alcedis GmbH, Amarex Clinical Research
**PRI-724**	Inhibitor of TCF/β-catenin transcription complex	Phase 1 and 2 NCT01606579	Advanced myeloid malignancies	inVentiv Health Clinical; Prism Pharma Co., Ltd.
**PRI-724**	Inhibitor of TCF/β-catenin transcription complex	Phase 1 NCT01302405	Advanced solid tumors	inVentiv Health Clinical; Prism Pharma Co., Ltd.

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
