# Peer review of "Wnt Pathway: An Integral Hub for Developmental and Oncogenic Signaling Networks"

_ijms, 2020, doi:10.3390/ijms21218018_

Round 1
Reviewer 1 Report
This is a fine review; a few suggestions for improvement in the context of a minor revision. First, I'm not sure that "miscues" is the best word to use in a table describing abnormal Wnt signaling components/regulation. Second, the section on inhibitors of the degradation complex seems to actually be about reagents that end up increasing beta-catenin destruction, so it should probably be termed regulators of the complex. Third, some work suggests inhibiting Wnt in established colon cancer may promote metastasis in some cases and other work suggests that abnormal very high upregulation of the pathway can induce apoptosis. So, it is more complex than simply "less is better" in all cases.
Author Response
RESPONSE TO REVIEWER 1 COMMENTS
This is a fine review; a few suggestions for improvement in the context of a minor revision.
Point 1: First, I'm not sure that "miscues" is the best word to use in a table describing abnormal Wnt signaling components/regulation.
Response 1: Thank you for the suggestion. We have now replaced the word “miscues” with “dysregulation” in Table 1 on page 5.
Point 2: Second, the section on inhibitors of the degradation complex seems to actually be about reagents that end up increasing beta-catenin destruction, so it should probably be termed regulators of the complex.
Response 2: Thank you for the feedback. We have renamed the subheading as the “Regulators of β-catenin destruction complex” on page 12.
Point 3: Third, some work suggests inhibiting Wnt in established colon cancer may promote metastasis in some cases and other work suggests that abnormal very high upregulation of the pathway can induce apoptosis. So, it is more complex than simply "less is better" in all cases.
Response 3: We agree with this point and have now better accounted for this level of complexity in the manuscript. We include additional references that better explain how, for example, some Wnt ligands can act as either tumor suppressors or oncogenes in various contexts on page 9.
Reviewer 2 Report
Sharma and Pruitt present an excellent review of the roles of Wnt signaling in cancer. The article starts will a succinct and well-illustrated overview of Wnt signaling pathways and their interactions with other cell signaling systems. This is followed by a detailed presentation of the available information about the effects of genetic abnormalities in Wnt signaling on the development of cancers of various localizations. Especially interesting is the chapter dedicated to role of Wnt pathways in the regulation of anti-cancer immunity. The article is concluded by the overview of available inhibitors of Notch signaling and their potential in cancer treatment. The authors critically discussed a large volume of available literature. The review will be very useful for biomedical researchers dealing with Wnt signaling.
Author Response
RESPONSE TO REVIEWER 2 COMMENTS
This is a fine review; a few suggestions for improvement in the context of a minor revision.
Point 1: Sharma and Pruitt present an excellent review of the roles of Wnt signaling in cancer. The article starts will a succinct and well-illustrated overview of Wnt signaling pathways and their interactions with other cell signaling systems. This is followed by a detailed presentation of the available information about the effects of genetic abnormalities in Wnt signaling on the development of cancers of various localizations. Especially interesting is the chapter dedicated to role of Wnt pathways in the regulation of anti-cancer immunity. The article is concluded by the overview of available inhibitors of Notch signaling and their potential in cancer treatment. The authors critically discussed a large volume of available literature. The review will be very useful for biomedical researchers dealing with Wnt signaling.
Response 1: We really appreciate you taking the time to review the article. We have incorporated a number of minor edits throughout the review now.
Reviewer 3 Report
I found overall the paper to be well organized and written.
Revisions include:
1) How this review article differs from the already published review articles about wnt signalling? A brief statement about that is required.
2) It would be interesting to read about the current problems and solutions targeting the wnt pathway
3) I found the conclusions to be very vague. A proper conclusion highlighting the progress made so far and what is expected in the next decade is needed.
4) Structures of the small molecule inhibitors.
Author Response
RESPONSE TO REVIEWER 3 COMMENTS
I found overall the paper to be well organized and written.
Point 1: How this review article differs from the already published review articles about wnt signalling? A brief statement about that is required.
Response 1: This review differs from previously published articles in two key ways. First, it highlights seminal historic papers providing the reader with an appreciation of how the field has developed. However, it also highlights very recent reports describing the role of DVL post-translational regulation that acts as a new mode of Wnt pathway regulation.
Point 2: It would be interesting to read about the current problems and solutions targeting the wnt pathway
Response 2: We now point the reader to other references which provide a very detailed description of the efficacy and safety outcomes for Wnt pathway inhibitors on page 10 – PMID: 31792354, 26306903, 24981364.
Point 3: I found the conclusions to be very vague. A proper conclusion highlighting the progress made so far and what is expected in the next decade is needed.
Response 3: We have further expanded this section and stated more concrete conclusions on page 14.
Point 4: Structures of the small molecule inhibitors.
Response 4: This is an excellent suggestion. Since our knowledge on the subject of structural biology is limited at the time, we have included 4 new references for our readers which nicely illustrate the structure of Wnt pathway inhibitors page 10. Some of the references now included in the revised paper are PMID: 24981364, 28120389, 29107427, 26522946, 31396368.
Reviewer 4 Report
The review article entitled “Wnt pathway: an integral……………network” by Monica Sharma and Kevin Pruitt summarizes the role of canonical and non-canonical Wnt pathways in developmental disorders and other diseases including cancers. Although the article is well written, it needs a minor but thorough edits by avoiding frequent use of ‘which’. My common comments are listed below:
- Panel A of the figure 1 showing canonical Wnt pathway is a little difficult to understand as it lacks arrows to link signaling events and somehow lacks a proper representation of Wnt-dependent fate of b-Catenin as it appears to be a separate component of the destruction complex. A better representation of this panel is suggested.
- The subheading “Wnt Pathway in development and diseases” is a little confusing. I suggest authors to create a separate subheadings for ‘Wnt Pathway in developmental disorders’ and ‘Wnt pathway in other diseases’.
- Difficult to understand why line numbers 255-257 and 259-263 are specifically shown in bold fonts.
- Please be consistent for writing G-proteins, c-myc etc. in an uniform format throughout the manuscript.
Author Response
RESPONSE TO REVIEWER 4 COMMENTS
The review article entitled “Wnt pathway: an integral……………network” by Monica Sharma and Kevin Pruitt summarizes the role of canonical and non-canonical Wnt pathways in developmental disorders and other diseases including cancers. Although the article is well written, it needs a minor but thorough edits by avoiding frequent use of ‘which’. My common comments are listed below:
Response: Thank you for this feedback. We have decreased the use of “which” and have incorporated a number of minor edits.
Point 1: Panel A of the figure 1 showing canonical Wnt pathway is a little difficult to understand as it lacks arrows to link signaling events and somehow lacks a proper representation of Wnt-dependent fate of b-Catenin as it appears to be a separate component of the destruction complex. A better representation of this panel is suggested.
Response 1: Thank you for the suggestion. We have rearranged Figure-1 and included arrows to represent the flow of canonical Wnt pathway. Please see the revised Figure-1 on page 3.
Point 2: The subheading “Wnt Pathway in development and diseases” is a little confusing. I suggest authors to create a separate subheadings for ‘Wnt Pathway in developmental disorders’ and ‘Wnt pathway in other diseases’.
Response 2: We appreciate the feedback. We have renamed the title as “Wnt pathway in human diseases”. Please see the changes on page 4.
Point 3: Difficult to understand why line numbers 255-257 and 259-263 are specifically shown in bold fonts.
Response 3: Thanks for the catching this. The bold font has now been removed from these lines.
Point 4: Please be consistent for writing G-proteins, c-myc etc. in an uniform format throughout the manuscript.
Response 4: We have made sure that we consistently use the same format for G-proteins, c-myc throughout the manuscript.